# Dynamic Capabilities and Firm Performance in the High-Tech Industry: Quadratic and Moderating Effects under Differing Ambidexterity Levels

**Michael Yao-Ping Peng [1] [iD], Zhaohua Zhang [2],\*, Hsin-Yi Yen [3],\* and Shu-Mi Yang [4]**

[1]  School of Management, Xi'an University of Architecture and Technology, Xi'an 710055, China
[2]  School of Humanities, Jinan University, Zhuhai 519070, China
[3]  Department of International Business, Providence University, Taichung 43301, Taiwan
[4]  Department of Business Administration, Ling Tung University, Taichung 408, Taiwan
\*  Correspondence: tzngg@jnu.edu.cn (Z.Z.); s8817007@gmail.com (H.-Y.Y.)

**Abstract:** This study uses the perspectives of dynamic capabilities and ambidexterity to investigate the direct effect of the development of an organization's explorative and exploitative capabilities on organizational tensions and performance. We employed a sample of high-tech Taiwanese firms to test our hypotheses and surveyed the informants' knowledge about their companies. We sent out 1000 questionnaires and received 234 valid responses, yielding a 23.4% effective response rate. The results also indicated that the consideration of incorporating balanced and combined dimension ambidexterity would benefit high-tech firms and help them facilitate higher performance. In summary, based on the results of previous studies, this study divided dynamic capabilities into exploitation capabilities and exploration capabilities, and divided ambidexterity into combined and balanced dimensions, so as to redefine the relationship between dynamic capabilities, ambidexterity and organizational performance from the perspective of tension, thereby enhancing the connotations of dynamic theory.

**Keywords:** ambidexterity; dynamic capability; high-tech industry; organizational tensions

## 1. Introduction

In research on enterprise resources and capacity accumulation, different theories have produced different interpretations of the focal issues [1–4]. Most scholars have emphasized the importance of dynamic capabilities (DCs) and noted that significant changes will take place in the direction and path of capabilities over time, so analyses of firm performance should focus on exploring capability development rather than resource development [2–5]. The dynamic capability view holds that, in a dynamic and changing environment, firms should have adequate flexibility and adaptability to deal with the challenges they face [2] and develop the capabilities needed to do so by identifying the best practices. In addition to the research that examined how to develop the capabilities [2,5] and influence of these practices on organizational performance and survival [6], the relationship between dynamic capabilities and performance still remains uncertain.

Zott [7] noted that even minor differences in the capabilities of firms would produce differences in their performance, as also found by Adner and Helfat [6]. This result is also supported by the research results of Helfat and Peteraf [5] and O'Reilly and Tushman [8]. Peng and Lin [9] adopted the claims of Fainshmidt, Pezeshkan, Frazier, Nair, and Markowski [10] and indicated that most of the previous studies discussed explorative and exploitative capabilities to measure DC constructs [2,5,11–15].

Based on the existing literature and arguments, this study aims to fill several critical knowledge gaps and provide some research contributions. First, the negative effect of DCs still remains

under-researched, as these effects are considered a "dark-side" issue [16]. Because of a lack of resources and a restricted management scope, some scholars have considered the relationship between explorative and exploitative capabilities to be a trade-off relationship [17], which means that investment in one capability may inhibit the development of the other [11,12,14] and thereby generate organizational tension. Although most studies point out the positive influence of explorative or exploitative capabilities on performance, there are still studies proposing that nonlinear or even negative relationships exist between dynamic capabilities and performance [12]. According to the tension-based view, conflicting forces within organizations are often hidden and pull on one another, eventually leading to organizational instability [18,19]. To facilitate organizational sustainability, tensions related to DCs are worth exploring [20]. In order to explore the "dark side" of DCs, it is necessary to clarify the relationships between DCs, as well as organizational tensions and performance. In addition to making theoretical contributions to the research gap of the "dark side" of DCs, this study also explores the non-linear relationship between DCs and organizational performance.

Second, earlier studies often regarded the trade-offs between these two capabilities as insurmountable, but more recent research describes ambidexterity that is capable of simultaneously exploiting existing competencies and exploring new opportunities [11,14,21–25]. Many relevant studies have examined ambidexterity as a way to combine explorative and exploitative capabilities in ways that positively impact firm performance [14,15,25,26]. On the other hand, promoting performance growth by enhancing one's dynamic capabilities is a good practice in market competition. Giroud and Mueller [27] studied this issue from the perspective of corporate governance, pointing out that the fact that firms with good corporate governance have better performance on average is well established. However, these studies have left a critical gap in terms of whether negative factors (organizational tensions) within the organization will disappear even if ambidexterity improves organizational performance [14,22,28,29]. Although Peng and Lin [9] explored the relationship between ambidexterity and organizational tension, they produced a similar research gap—whether or not reducing organizational tensions will enhance performance. Therefore, this study aims to explore dynamic development among ambidexterity, organizational tensions, and performance.

Third, based on the results of previous works, this study divides dynamic capabilities into exploitation and exploration capabilities, and ambidexterity into the combined dimension of ambidexterity (CD) and the balanced dimension of ambidexterity (BD) [14,21], so as to redefine the relationships among dynamic capabilities, ambidexterity and organizational performance from the perspective of tension, thereby enhancing the application of dynamic theory.

## 2. Literature Review and Hypotheses Development

### 2.1. Dynamic Capabilities and Organizational Performance

Due to improvements in the competitive advantage of many enterprises, Teece and Pisano [30] proposed the concept of dynamic capabilities (DCs), drawing from a resource-based view [1]. In long-term competition, the continual development of dynamic capabilities is needed to maintain a firm's competitive advantage [2,31]. Through these processes, firms constantly integrate, reconfigure, renew, and recreate resources and capabilities in response to the changing environment to attain and sustain competitive advantage [1,2,31]. Fainshmidt et al. [10] combine the ambidexterity of the dynamic capability theory of the past with related research, and this research indicates that the relevance of higher-order dynamic capabilities to organizational performance will be stronger than that of lower-order dynamic capabilities. This means that dynamic capabilities will be made up of different types of measurement variables. Conversely, the majority of studies have been focused on exploitation and exploration capabilities [8,9,11,14,15].

The essence of explorative capability emphasizes the creation of something beyond the existing knowledge of an organization, such as the development of new knowledge, by using new methods to experiment with technologies, business processes, or markets and to search for new organizational

norms, practices, and systems [32]. In a study on the internationalization process of enterprises, Prange and Verdier [13] indicated that explorative capabilities are best seen in firms that dynamically make use of value-adding or disruptive capabilities in order to achieve new and innovative competitive advantages [11]. They also noted that a firm's destructive capabilities can increase the tendency for organizations to engage in structure-destroying changes, letting them overcome path-dependence and inertia to expedite organizational growth. Accordingly, explorative capabilities can create new products and services, as well as develop new markets [23] and also enable organizations to seek more appropriate structures [14].

However, the single-minded pursuit of explorative capabilities can be limiting, as firms that engage in explorative processes take advantage of expenditures or resources that originally belonged to exploitative processes, and this can lead to substantial experimental costs and even losses [11,12]. Failure often impels organizations to adopt explorative strategies, as the dynamics of failure will sink organizations into a cycle of unusual experiments, change, and innovation [11,12]. With financial resources invested in rapid expansion, new routines, and/or adaptation, firms engaged in explorative capabilities may fail to garner direct profits from continuous profit sources. The establishment of explorative capabilities and the implementation of such activities often require more time than exploitive capabilities and their related activities. Moreover, the former must bear the risks and costs associated with greater uncertainties [33]—e.g., firms must develop new products directed at customer demand and new markets, thereby siphoning capital, resources, and manpower from other projects [14,21]. The development of new products often belongs to a long-term orientation. Firms can achieve significant revenue through the sale of new products in the initial stages of the product's life cycle. However, without fixed revenue from existing product lines and markets, this may result in higher development and operating costs than sales revenue in later stages of the product's life cycle [9,13]. Therefore, we hypothesize the following:

**Hypothesis 1 (H1).** *Explorative capabilities have an inverted U-shaped relationship with organizational performance.*

Exploitative capabilities are dynamic capabilities, involving activities such as path-dependent learning and knowledge storage [11]. Firms tend to stress the development of existing markets, avoiding extending to new ones until they accumulate adequate capabilities [11,12]. This step-by-step approach can reduce the uncertainties that exist in relation to exploration and experiments, and thus improve the chances that such firms will survive [13]. Slater and Narver [34] proposed that firms that engage in continuous learning will tend to track and respond to consumer demands, thereby identifying and capturing market opportunities while providing suitable products to enhance profitability, sales growth, and customer retention. The accumulation of experience and lessons that takes place in this way can make enterprises more aware of how to avoid repeating mistakes, how to reduce production and transaction costs, and how to strengthen employees' capabilities of mutual understanding, as well as problem-coordination and problem-solving [35]. If an organization merely engages in exploitation and excludes exploration, then it might suffer due to changes in technology or customer preferences [2,11,12,14,15]. Moreover, when an organization succeeds, its attention often shifts from explorative to exploitative activities, since companies in a well-defined field tend to develop capabilities that further enhance their specific competences, while the opportunity cost of exploration also increases, as noted in Prange and Verdier [13]. However, firms that do not work to achieve organizational growth by applying their explorative capabilities risk declining over the long term. Therefore, we hypothesize the following:

**Hypothesis 2 (H2).** *Exploitative capabilities show a U-shaped relationship with organizational performance.*

## 2.2. The Tensions View

Previous studies have pointed out that explorative and exploitative capabilities are closely bound to organizational survival and growth [2,5,11–14]. Although explorative and exploitative capabilities automatically transfer the related efforts into superior performance [15], this does not mean a firm can overcome, integrate, or reduce existing contradiction and conflict [19]. This is a gap in the literature, since previous studies fail to consider such "dark-side issues". In addition, Li [36,37] discussed the role of mutual monitoring in corporate governance. Li pointed out that information asymmetry exists in the organization among senior managers, who may have different understandings of resource input of competence and generate different tensions derived from the authority, credibility, and influence of team members.

Moreover, due to their limited resources, companies play favorites when allocating resources. Thus, pulling from both ends of the explorative and exploitative continuum causes organizational tensions [12,14,19,22]. This study refers to Andriopoulos and Lewis [33], who divided the tensions between explorative and exploitative capabilities into strategic intent tension (profit emphasis vs. breakthrough emphasis), customer orientation tension (tight coupling vs. loose coupling), and personal driver tension (discipline vs. passion). These divisions are explained in more detail below.

*Strategic intent* is a firm's reason for being, and related tensions arise from the need to emphasize both profit and breakthroughs. As He and Wong [14] and Piao and Zajac [12] explained, exploitation and exploration seek opposing goals: stable revenues that enable higher mean performance and frame breaking opportunities that foster greater performance variation. Numerous firms that emphasize profits are infused with the spirit of conservatism and stress the value of repeat clients and efficiency. Moreover, they tend to focus on the development of exploitative capabilities. In contrast, high-tech firms that attach importance to breakthroughs will strive to find new opportunities and technologies to enhance new product development to maintain their innovative position in the market [33]. However, in order to build barriers from imitation, these high-tech firms must continue to invest resources in research and development in order to prevent products from being outdated or imitated by competitors. Relatively speaking, investing too many resources in explorative capabilities may help to develop new products, but it may also lead to the risk of an excessive inventory of existing products, which may result in an increase in operating costs and inventory costs [31]. This may further cause imbalanced tensions.

*Customer orientation* attaches much importance to activities that help understand customer demands, needs, and behaviors. A company with this orientation shows an earnest learning attitude, such as through responding to customer demands and obtaining and absorbing related information [38]. Tensions related to customer orientation are based on whether the company needs to be tightly or loosely coupled to the client during projects. Tight coupling to the customer can be a double-edged sword; although close relation often provides contributions into the current market and yields higher customer satisfaction and loyalty [38–40], it may increase the relational cost to maintain existing relationships with customers [41]. In contrast, being loosely coupled is conducive to the exploration of new products or markets, thereby enabling high-tech firms to identify and handle future opportunities and customer demands. For a high-tech firm, the resources invested in explorative capabilities will be greater than those invested in exploitative capabilities, and investments in maintaining customer relationships will be relatively reduced. Although being loosely coupled makes it easier for firms to achieve an innovative status and meet customer needs, the lack of stable customer sources may lead to the risk of customers transferring to different firms [33,40].

Accordingly, the simultaneous development of explorative and exploitative capabilities may affect a firm's choice to connect with customers, and thereby produce imbalances and tensions.

*Personal drivers* refer to employees' personality traits, which are broadly divided into discipline and passion. If the staff is discipline driven, they work in accordance with the development processes, targets, and roles defined by the firm to perform organizational tasks. Conversely, if employees are passion driven, they seek individual performance and challenges, and the firm is thus able to stimulate the creativity of its knowledge workers by increasing this motivation [33]. Discipline-oriented

employees are engaged in standard routines and emphasize improving work efficiency and productivity rather than pursuing efficiency and innovation [42]. High-tech firms over-investing in exploitative capabilities will pursue a high degree of standardization, which may lead to organizational alienation and rigidity [43,44], and thus produce a lack of countermeasures to market change [45]. Relatively speaking, high-tech firms that invest excessively in explorative capabilities will pursue continuing technological advances. Although employees have a high degree of flexibility and freedom, there are tremendous pressures from innovation responsibility systems and industrialized work situations [46], which may result in a high turnover and a loss of focus on organizational goals [44,45]. Therefore, excessive emphasis on either discipline may lead employees to engage in a single explorative or exploitative activity, which could produce increased tensions [19]. These ideas are described in the following hypotheses.

**Hypothesis 3 (H3).** *The strength of high-tech firms' explorative capabilities is positively associated with organizational tensions.*

**Hypothesis 4 (H4).** *The strength of high-tech firms' exploitative capabilities is positively associated with organizational tensions.*

Organizations are subjected to tensions caused by conflicts in different fields [19]. The tension between exploitation and exploration capabilities means that firms will inevitably suffer in some ways no matter what kind of measures they take in this regard [14,21,33]. A favorable environment will result in structural inertia and reduce the adaptability to cope with environmental changes. Furthermore, any new scheme will reduce the existing speed of improving current capabilities [12,14,21]. In the current fields within which a firm operates, a failure to engage in exploration activities can collapse conventional routines and prevent the company from making up for its losses in this field with its successes in others [11]. This is summarized in the following hypothesis.

**Hypothesis 5 (H5).** *Organizational tensions are negatively associated with organizational performance.*

### 2.3. Effects of Ambidexterity on Organizational Performance and Tensions

While an organization facing fundamental problems devotes itself to using its exploitative capabilities to ensure its continued viability, it also needs to apply some organizational energy to developing its explorative capabilities, in order to survive into the future [12,14,19]. Ebben and Johnson [47] examined a sample of 300 small enterprises and found that those pursuing efficiency and flexibility had worse performance than those using a single and concentrated strategy. Ambidexterity at the organizational level can be defined as the ability to reconcile two opposite strategies (for example, simultaneously pursuing both exploration and exploitation) within the same firm [11,14,21–25,48]. Ambidexterity can also be considered good corporate governance. Li [36,37] indicated that mutual monitoring may arise in the normal methods associated with effective management.

Scholars have different views on ambidexterity, which have led to differences in their research results. He and Wong [14] and Cao et al. [21] divided ambidexterity into two different levels, one being the balanced dimension of ambidexterity (BD), which balances the relationship between exploration and exploitation [12], and the other being the combined dimension of ambidexterity (CD), which represents the combined strength of exploration and exploitation [22]. Cao et al. [21] and Gibson and Birkinshaw [22] attained different results in their studies. This difference is because although both BD and CD have positive contributions to firm performance, BD can reduce its impact on performance due to the damage to exploration cause by over-engagement in exploitation, while CD can derive complementary resources by leveraging exploration and exploitation, thereby enhancing firm performance [28]. Adopting BD and CD at the same time can help firms completely absorb

existing knowledge and resources and combine these with new knowledge, resources, and capabilities, thereby generating synergistic performance benefits.

The concept of ambidexterity in this context refers to the interaction effect of explorative and exploitative capabilities. Thus, these two concepts are not perceived as being in competition with each other and do not have a trade-off relationship, but instead are seen as having a complementary relationship [14,21,22,28]. In other words, exploitative capabilities can help firms explore new knowledge and develop the resources needed to support new products and markets (Cao et al., 2009). On the other hand, mastering explorative processes can enhance a firm's capacity for exploitative activities. Andriopoulos and Lewis [33] pointed out in their research on the tension between exploitation and exploration activities that CD represents situational ambidexterity and stressed that activities in opposite relationships are mutually dependent. By proposing an integration strategy, they effectively managed strategic intention, customer orientation, personnel driven factors, and other tension factors [19]. For example, organizations with situational ambidexterity stress the synergy between benefits and breakthroughs and foster the perception of greater conflicts. This is because, when firms internalize external knowledge and resources by using their explorative capabilities, they thus extend their own competences, so more effective routines and processes are utilized in larger economies of scale. Ambidexterity can, therefore, leverage the synergy between new opportunities and the limitations of existing routines and knowledge, in a process of improvisation that was described by Miner et al. [49] and Mathias et al. [19]. We thus hypothesize the following:

**Hypothesis 6 (H6).** *The strength of CD will positively correlate with organizational performance.*

**Hypothesis 7 (H7).** *The strength of CD will negatively correlate with organizational tensions.*

BD represents the relative strength of the exploitation and exploration capabilities or activities of firms [21]. If the relative value is large, the relationship between the two must be imbalanced, and thus the organizational tension will be intensified [19], which will further enhance the performance risks and harm organizational development [12,14]. In contrast, if the relative value (absolute difference) is small or equal, the two kinds of capabilities must be closely fitted [12,14,21]. The performance risk and firm engagement in exploitation activities will be greater than that in exploration activities, so firms may be trapped with a risk of obsolescence. In other words, firms with such risks will tend to pursue success in the short term by virtue of their existing products and markets. However, this success is short-lived, as they eventually have to face up to changes in the market and technology [28]. Therefore, ambidexterity that results from the balance between exploitation and exploration capabilities or activities will help firms control the profits gained from existing products and markets, thereby reducing the performance risks caused by over-engagement in each of these.

In addition, Andriopoulos and Lewis [33] pointed out that BD differs from CD [29,50]. By virtue of adopting a differentiation strategy, firms are encouraged to develop different structures to engage in conflicted, opposing activities, in order to reduce organizational tension [19]. For example, according to different strategic objectives, firms can enrich conventional, profitable projects and high-risk breakthrough projects to diversify their investment portfolios, so firms can leverage their existing strengths and accept more risks to increase future market opportunities [51]. In terms of customer orientation, a differentiation strategy also refers to the separation of time. Firms will separate organizational tension based on an iterative process that moves between the constraints and freedoms of projects and thus learn about customer expectations and the market's prospects in order to gradually lose their initial constraints, before considering new fields of opportunities through brainstorming [39]. Finally, BD can help firms make good use of the separation of time and space to divide the specific tasks and time periods of different projects, to better deal with employees who have different motivations and abilities, with knowledge workers better able to leverage their discipline and passion [33]. To sum up, this study concludes with the following hypotheses and research structure.

**Hypothesis 8 (H8).** *The strength of absolute difference (meaning lower BD) will negatively correlate with organizational performance.*

**Hypothesis 9 (H9).** *The strength of absolute difference (meaning lower BD) will positively correlate with organizational tensions.*

Based on the above hypotheses and literature discussions, this study proposes the following research framework, as shown in Figure 1.

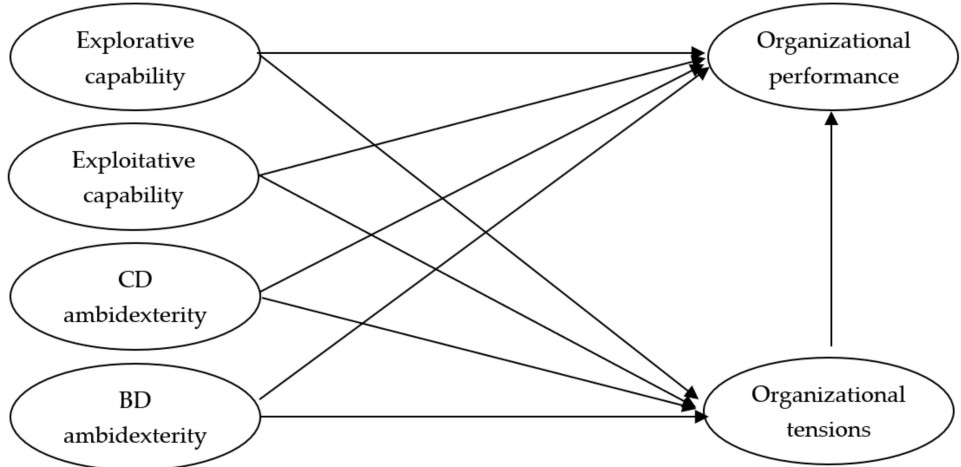

**Figure 1.** Conceptual framework.

## 3. Methodology

### 3.1. Sampling

The sampling method involved the dissemination of questionnaires to high-tech manufacturers in Taiwan, with the target audience being the people in charge of the companies or the supervisors most familiar with internal operations, as such personnel have mastered most of their companies' operating conditions and are more familiar with internal operating practices. Top managers were chosen as the respondents, as it is assumed that they are very familiar with their companies' operations. We sent out 1000 questionnaires and received 237 completed responses, giving a 23.7% response rate. After eliminating three invalid questionnaires, there were a total 234 valid responses, for a 23.4% effective response rate. Most respondents (88.7%) were located in the electronics, electric appliance, metal, machinery, plastics, textiles, and automobile manufacturing industries; 30.7% had been established for 3–10 years, 34.5% for 11–20 years, 19.7% for 21–30 years, and 15.1% for 31 years or more.

We compared early respondents with late respondents from the sampling frame by running an independent-samples *t*-test on all included items. The results indicate no significant differences, so we are not concerned about a non-response bias.

### 3.2. Measures

The questionnaire variables in this study were chiefly developed based on scales in the previous literature. Except for firm size and age, all questions were answered using a seven-point Likert scale. The five questions on the explorative capability scale (e.g., introducing a new generation of products) and four questions on the exploitative capability scale (e.g., improving the quality of existing products) were taken from He and Wong [14], Lubatkin et al. [24], Menguc and Auh [25], and Cao et al. [21]. We asked the respondents to state how their firms divided attention and resources between exploitative and explorative activities in the last three years.

Following Gibson and Birkinshaw [22], the concept of CD used in this work, a multiplicative term of explorative and exploitative capabilities, conforms to the theoretical conceptualization of ambidexterity [29]. The BD score is mainly the absolute value of the difference between the exploitation capability and exploration capability. A smaller absolute difference represents a higher degree of BD.

Following Andriopoulos and Lewis [33] and Groysberg and Lee [52], three dimensions were used to construct organizational tension: four items for strategic intent, three items for customer orientation, and four items for personal drivers. Organizational tension was measured by binominal semantic differential scales, as the endpoints consisted of bipolar activities, with assigned values of one and seven [53]. This study used a binominal scale to understand whether the tensions within organizations were balanced or unbalanced [9]. Taking items in customer orientation as examples, relationship orientation vs. transaction orientation indicates two ends of the item, so the excessively concentrated relationship orientation is one and, on the contrary, the transaction orientation is seven. The nearer the score is to four, the smaller the organizational tensions will be; the nearer the score is to one or seven, the bigger the organizational tensions will be.

Firm performance is a complex construct, and this study used perceived measures to assess this construct in terms of organizational effectiveness (three items), growth/market share (three items), and profitability (three items), based on an established reflective scale [24,34,54]. However, scholars believe that there are some similarities between growth and market share [34], so this study classifies both together as one of the measurement variables to measure performance. The senior managers surveyed in this work were thus asked to assess their firms' performance relative to their competitors' performance for the past three years. This study also controlled two other variables that might affect the model: firm size and age [21]. Based on Dang and Li's [55] arguments, firm size is a more important fundamental firm characteristic than other control variables. Therefore, it is more likely that a correlation exists between firm size to firm performance. To manage this fixed effect, this study uses the number of employees to measure firm size.

## 4. Results

### 4.1. Reliability and Validity

The descriptive statistics for the scales are summarized in Table 1. Confirmatory factor analysis and AMOS 22.0 were used to measure the reliability and validity of the scale. The construct validity of the scale was verified in terms of its convergent and discriminant validity. According to Hair, Black, Babin, Anderson, and Tatham [56], the evaluation standards for convergent validity are a standardized factor loading higher than 0.5, an average variance extracted (AVE) higher than 0.5, and a composite reliability (CR) higher than 0.7. The evaluation standard for discriminant validity is the square root of the AVE for one dimension greater than the correlation coefficient with any other dimension(s). As Table 1 indicates, all three criteria for convergent validity were met, and the correlation coefficients were all less than the square root of the AVE within one dimension, suggesting that each dimension in this study had good discriminant validity.

**Table 1.** Statistics.

| Measure | Means | SD | 1 | 2 | 3 | 4 | 5 | 6 | 7 | 8 | 9 | 10 | 11 | 12 |
|---|---|---|---|---|---|---|---|---|---|---|---|---|---|---|
| 1. Explorative | 5.05 | 1.08 | 0.79 | | | | | | | | | | | |
| 2. Exploitative | 5.04 | 1.11 | 0.70 ** | 0.76 | | | | | | | | | | |
| 3. Strategic intent | 3.68 | 1.33 | −0.09 | −0.13 | 0.77 | | | | | | | | | |
| 4. customer orientation | 3.74 | 1.33 | −0.10 | −0.17 ** | 0.81 ** | 0.76 | | | | | | | | |
| 5. Personal driver | 3.82 | 1.16 | −0.11 | −0.21 ** | 0.76 ** | 0.71 ** | 0.81 | | | | | | | |
| 6. CD | 0.65 | 0.69 | 0.01 | −0.12 | −0.18 ** | −0.18 ** | −0.11 | – | | | | | | |

**Table 1.** *Cont.*

| Measure | Means | SD | 1 | 2 | 3 | 4 | 5 | 6 | 7 | 8 | 9 | 10 | 11 | 12 |
|---|---|---|---|---|---|---|---|---|---|---|---|---|---|---|
| 7. BD | 0.55 | 1.14 | −0.23 ** | −0.14 * | 0.18 ** | 0.19 ** | 0.10 | −0.32 ** | – | | | | | |
| 8. Organizational effectiveness | 4.93 | 1.16 | 0.12 | 0.14 * | 0.16 * | 0.14 * | 0.14 * | −0.09 | −0.13 | 0.81 | | | | |
| 9. Growth and share | 4.96 | 1.06 | 0.07 | 0.08 | 0.22 ** | 0.20 ** | 0.14 * | −00.18 ** | −0.04 | 0.78 ** | 0.80 | | | |
| 10. Profitability | 4.79 | 0.97 | 0.08 | 0.07 | 0.19 ** | 0.15 * | 0.11 | −00.15 * | −0.06 | 0.53 ** | 0.66 ** | 0.80 | | |
| 11. Firm size | 2.80 | 1.39 | 0.09 | 0.06 | −0.10 | −0.05 | −0.13 | 0.09 | 0.00 | −0.05 | −0.01 | 0.04 | – | |
| 12. Firm age | 3.36 | 1.23 | 0.01 | 0.09 | 0.05 | 0.00 | −0.05 | 0.12 | −0.06 | 0.00 | 0.02 | −0.05 | 0.30 ** | - |
| AVE | | | 0.62 | 0.57 | 0.59 | 0.58 | 0.66 | – | – | 0.66 | 0.66 | 0.64 | – | - |
| CR | | | 0.89 | 0.85 | 0.85 | 0.80 | 0.88 | – | – | 0.85 | 0.84 | 0.84 | – | - |

* $p < 0.05$; ** $p < 0.01$.

## 4.2. Hypotheses Testing

This study used two approaches to examine the hypotheses. First, a hierarchical regression model was employed to demonstrate the main effects, namely the effect of explorative and exploitative capabilities on organizational performance and tensions. Second, to assess the effects of BD and CD on the organizational performance and tensions, multiple linear regression analyses were conducted separately. A formal measure used to test for the presence of multicollinearity in regression analysis is an assessment of the variance inflation factor (VIF) for each independent variable [56]. The maximum VIF for the independent variables in this study is 3.834, indicating that multicollinearity is not a major problem in the analysis. Li [57] pointed out in the study that performance in different periods may be influenced by dependent variables, which may further generate potential endogeneity issues. However, this study required the respondents to evaluate the degree of development at a specific time point, and the questions were also designed in the questionnaire to avoid tautology. Thus, this study did not conduct endogeneity tests for the measurement variables.

Table 2 shows the results of the multiple linear regression analyses. Inconsistent with the hypotheses, the study results demonstrate non-significant relationships among the explorative capabilities, exploitative capabilities, and organizational tensions in Models 1 and 2. Thus, H3 ($\beta = 0.016$ and $−0.044$, $p > 0.1$) and H4 ($\beta = 0.107$ and $0.044$, $p > 0.1$) were not supported. Since we measured CD as the multiplication of the explorative/exploitative capability constructs and examined the influence of squared explorative and exploitative capabilities on organizational performance, we acknowledge that this measure may suffer from multicollinearity. To minimize this concern in our analyses, we mean-centered the constructs of the exploitative and explorative capabilities before deriving the measurement of ambidexterity (Aiken and West, 1991). As presented in Table 2, exploitative capabilities ($\beta = 0.202$ and $0.179$, $p < 0.001$) and explorative capabilities ($\beta = −0.398$ and $−0.266$, $p < 0.001$) show a significant, non-linear relationship with the organizational performance in Models 3 and 4, while organizational tensions ($\beta = −0.220$ and $−0.260$, $p < 0.001$) have a negative relationship with organizational performance. These results provide support for H1, H2, and H5.

Further, following Cao et al. [21], who evaluated the effects of alternate ambidexterity variables in separate models, we first estimated the models in which only one of the two ambidexterity dimensions were entered. To study the organizational tensions and performance, Models 1 and 4 include the main effect of BD only, and Models 2 and 3 examine the main effect of CD only. In these models, we find that both BD ($\beta = −0.209$, $p < 0.01$; $\beta = 0.141$, $p < 0.001$) and CD ($\beta = 0.141$, $p < 0.001$; $\beta = −0.180$, $p < 0.01$) are significantly related to organizational tensions and performance, which is consistent with the results reported by Cao et al. [21] and He and Wong [14]. H6, H7, H8, and H9 were thus supported.

**Table 2.** Linear regression analyses.

| | Dependent Variable: Organizational Tensions | | Dependent Variable: Organizational Performance | |
|---|---|---|---|---|
| | **Model 1** | **Model 2** | **Model 3** | **Model 4** |
| Control variable | | | | |
| Firm size | 0.099 | 0.120 † | −0.007 | 0.000 |
| Firm age | −0.052 | −0.052 | 0.029 | 0.064 |
| Main Effect | | | | |
| Explorative capability | 0.016 | −0.044 | −0.091 | −0.273 *** |
| Exploitative capability | 0.107 | 0.044 | 0.257 *** | 0.504 *** |
| Explorative capability $^2$ | | | −0.398 *** | −0.266 *** |
| Exploitative capability $^2$ | | | 0.202 *** | 0.179 *** |
| Ambidexterity | | | | |
| CD | | −0.180 ** | 0.162 * | |
| BD | −0.209 ** | | | 0.141 *** |
| Organizational tensions | | | −0.220 *** | −0.260 *** |
| F-value | 2.449 * | 2.047 † | 15.089 *** | 17.587 *** |
| R$^2$ | 0.051 | 0.043 | 0.450 | 0.488 |
| Adj R$^2$ | 0.030 | 0.022 | 0.420 | 0.461 |
| DW | 1.954 | 1.967 | 1.474 | 1.626 |
| Max VIF | 3.249 | 2.881 | 3.834 | 3.484 |

† $p < 0.1$; * $p < 0.05$; ** $p < 0.01$; *** $p < 0.001$. $^2$ means squared.

### 4.3. Post Hoc Examination of Findings

To explore the patterns of firms with respect to their pursuit of BD or CD, we performed a post hoc cluster analysis. Applying the K-means algorithm, a four-group model provides the best fit for the data. Groups were constructed based on the firms' BD and CD (rated according to the results set out above) as low BD/CD, low BD/high CD, high BD/low CD, and high BD/CD. Table 3 plots the BD and CD levels for each of the four cluster centers and reports the univariate F-statistics that show the four groups differing significantly in their levels of BD and CD. By examining the level of tension (strategic intent, customer orientation, personal drivers) and performance (effectiveness, growth/share, and profitability) for each group of high-tech firms, we found that Group 4, containing most ambidextrous firms, reported the highest levels of subsequent performance. Some scholars use the term "dual orientation ambidextrous" [8], which facilitates the dynamic incorporating both BD and CD, and thus increases the awareness of opportunities to improve working efficiency and innovation. At the other extreme, Group 1 had a lower number of ambidextrous firms and those with the lowest performance.

**Table 3.** Results of the post hoc cluster analysis.

| Firms | N | Strategic Intent | Customer Orientation | Personal Drivers | Effective-Ness | Growth/Share | Profitability |
|---|---|---|---|---|---|---|---|
| Group 1 | 57 | 3.623 | 3.836 | 3.811 | 4.532 | 4.579 | 4.415 |
| Group 2 | 52 | 3.115 | 3.109 | 3.375 | 5.038 | 4.878 | 4.795 |
| Group 3 | 66 | 3.894 | 3.949 | 4.004 | 5.006 | 5.115 | 4.816 |
| Group 4 | 59 | 3.970 | 3.971 | 4.004 | 5.116 | 5.187 | 5.061 |
| *F*-value | 234 | 4.919 ** | 5.441 *** | 3.720 * | 3.089 * | 4.156 ** | 4.865 ** |

* $p < 0.05$; ** $p < 0.01$; *** $p < 0.001$.

As the firms in Group 1 (lacking ambidexterity) do not clearly know about their strategic orientation and tend to "muddle through," and they fail to connect problems and opportunities to resolve them. While ANOVA showed that firm tensions and performance ($F = 19.30$, $p < 0.001$) vary significantly across groups, the post hoc S-N-K (Student–Newman–Keuls) procedure indicated statistically insignificant differences between Groups 2 and 3. In other words, Group 2 and 3 firms, which were moderately ambidextrous but tended to favor either BD or CD, had moderate levels of firm tensions and performance. Firms in Group 2 (CD firms) possess highly refined routines that leverage clearly identified core strengths and focus on synergy-driven rents. In addition, firms in Group 3 (BD firms) are seeking opportunities to access new technologies, products, and markets, as well as pursuing successful commercialization and profit efficiency. The firms in Group 3 were similar to each other, yet significantly different from Groups 1 and 4. These results, therefore, provide additional empirical evidence for the theoretical logic underlying our hypotheses: high-tech firms with ambidexterity can better jointly pursue BD and CD and thereby attain a higher level of relative subsequent performance.

## 5. Conclusions

In uncertain and dynamic environments, in order to adapt to changes in the external environment and the modifications of enterprise strategies, organizations need to construct the necessary capabilities while also continuously changing their characteristics or else risk being pushed out of the market or industry [2–5,11]. A dynamic capability is part of an enterprise's competence, and its development is in line with the capability life cycle theory [5]. It is thus worth exploring how dynamic capabilities adapt to changes in the external environment through development and evolution, and what mechanisms guide enterprises to implement such activities. Previously, scholars held that explorative and exploitative capabilities had a positive influence on organizational performance by improving long-term performance and market share, reducing costs, enhancing flexibility, accelerating the development of new products, and so on [2,5,11–15]. However, due to some tensions between these capabilities, there may be a trade-off between them, which could have a negative impact on organizational performance [19,33]. By developing organizational competence in relation to ambidexterity, firms can better leverage their own resources and capabilities to determine the practices that can reduce the negative tension that arises due to the trade-off between explorative and exploitative capabilities, and thus enhance organizational performance.

The empirical results show that there is a curvilinear relationship between explorative capabilities (inverted U shape), exploitative capabilities (U shape), and organizational performance, which conforms to the arguments established in this study. This result means that, in the case of limited resources, a firm's continuous investment in the development of specific capabilities will have a trade-off effect on the development needs of other capabilities. Therefore, in this study, the concept of ambidexterity was adopted to discuss the relationship between the development of dynamic capabilities and the tension between explorative and exploitative capabilities. The empirical results show that both BD and CD can effectively reduce the negative influence of tension, thereby improving organizational performance. This result is consistent with the claims of Cao et al. (2009), Gibson and Birkinshaw [22], and Koryak et al. [28], which show that explorative and exploitative capabilities are mutually supportive, as each leverages the strengths of the other to complement each other's weaknesses. Organizational ambidexterity provides organizational members with a learning environment in which they can dynamically adjust their learning processes (i.e., both explorative and exploitative capabilities) [24] and better understand existing systems to develop new ideas and methods and improve productivity [11,14,23–25,58]. In other words, high exploitative capabilities can usually improve a firm's efficiency in exploring new knowledge and developing resources that are conducive to launching new products and expanding into new markets.

According to our findings and the above arguments, this study provides several contributions. First, this study showed the dark-side impact of DCs—influencing the operation and development of an organization means that even if organizational ambidexterity is promoted, the existence of tensions

cannot be changed. For the study of dynamic capabilities, we must first understand the negative effects of tensions and then discuss the relationship between DCs and performance. Second, this study not only strengthens the description of DCs in previous studies but also integrates ambidexterity measurement methods to provide more in-depth insights. Third, although explorative and exploitative capabilities are not positively related to organizational tensions, there are still many possibilities for the combination of ambidexterity and organizational tensions in the field of DCs, which could provide a direction for future research. For example, future researchers could explore the formation of tension at the individual level, based on the concept of corporate governance [27,36,37,59].

### 5.1. Practical Implications

The dynamic capabilities of a firm are a special kind of capability derived from a series of organizational routines, processes, and structures in long-term operation and management, which are the integration mechanisms of enterprise resources and a source of sustainable competitive advantages [2,8]. As Piao and Zajac [12] noted, if a firm merely concentrates on a single capability, it will be trapped in a vicious spiral [48]. If a firm is committed to developing its exploitative capabilities, it may tend to engage in highly homogenous activities and develop the routines that can support development and innovation in a relatively relaxed manner, thereby leading to "competence traps" [26]. In contrast, if a firm narrowly develops its explorative capabilities, it may overlook core competences and attempt to deny past failures during innovation. Therefore, a firm should possess enough structural flexibility to adapt to its environmental requirements, in order to adjust, integrate, and utilize scattered, conflicting activities in different organizations and allocate, reallocate, combine, and recombine its resources and assets. Specifically, this claim happens to coincide with balanced ambidexterity. The organization of ambidexterity follows the idea that an organization can simultaneously engage in two conflicting, dual, activities, capabilities, procedures, or orientations, which depend on internal calibration. This claim also confirms the importance of structural differentiation or dual structures. Organizations constructed with a dual structure [48] can give an organization the ability to handle knowledge integration problems and management produced by internal complexity and diversity, thereby promoting the organization to identify valuable knowledge and integrate related information.

However, how should managers establish organizational situation with ambidexterity? According to previous studies, and based on this study, it is suggested that an organizational situation with ambidexterity established by the top managers should have four characteristics: stretch, discipline, support, and trust. These four characteristics can be combined to construct two organizational dimensions: performance management (the combination of stretch and discipline), which stimulates members to deliver high-quality results and be responsible for their own actions, and social support (the combination of support and trust), which provides members with a sense of security and freedom in the process of carrying out a task. For example, a lack of social support will lead to lassitude. As members carry out tasks within a limited time (the essence of inhumanity) egoism and authoritative orientation will result in a high employee turnover rate, and thus make it difficult to realize ambidexterity. Similarly, strong social support without expectations for high performance will lead to a country club situation, where internal members will become immersed in a collegial environment and rely on others' help, thus making it difficult to improve their own potential.

Finally, top management teams should further develop a strong and convincing shared vision. For example, Li et al. [59], who developed their study from the perspective of corporate governance and compensation theory, justify the use of inside debt as an effective incentive that is superior to other solvency-contingent instruments (e.g., salaries, bonuses, reputation, and private benefits) because it renders managers sensitive not only to the firm's internal development but also to the firm's value in the event of organizational tensions. This is because a collective ambition will offer a future development path for the organization, and the organizational members with common values will give priority to explaining problems and reducing conflicts. When a strong shared vision becomes the main mechanism

of the organization, it will help the firm balance the allocation of resources required by explorative and exploitative activities through integration or synergy.

## 5.2. Research Limitations and Future Research Suggestions

This research is not without its limitations and thus should prompt further study. First, the sample frames used in this study were selected from high-tech industries, though this industry covers diverse products that have great differences between them in their production scales and resource investments. For example, independent software vendors and semi-conductor manufacturers have different demands on capabilities and resources. In addition, this study adopts firm size as the control variable. Lubatkin et al. [24] suggest that executives from larger firms should reconsider separating units to focus on either exploitation or exploration and should instead create units capable of pursuing both. Future studies, then, can test the specific products of the high-tech industry to analyze the differences between large and small firms. Second, this study failed to consider how to avoid common method variance when issuing questionnaires. Future studies should consider and avoid the occurrence of common method variance when carrying out research and experimental designs. Problems arising from this variance will affect not only the results of statistical analysis but also the processes and results of the methods and verification, meaning that relevant issues and the verification of common method variance should be studied further. Finally, we focused on selected constructs to develop our research framework. Although this strategy helped us maintain conceptual clarity and parsimony, we may have overlooked other variables. We encourage more research on the organizational, interfirm, and environmental determinants of organizational ambidexterity.

**Author Contributions:** Writing—original draft M.Y.-P.P.; Conceptualization, Z.Z.; Methodology, M.Y.-P.P. and H.-Y.Y.; Investigation, M.Y.-P.P., H.-Y.Y. and S.-M.Y.; Validation, M.Y.-P.P., Z.Z. and S.-M.Y.; Editing, H.-Y.Y.

**Funding:** This research received no external funding.

**Conflicts of Interest:** The authors declare no conflict of interest.

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
