# Peer review of "Dynamic Capabilities and Firm Performance in the High-Tech Industry: Quadratic and Moderating Effects under Differing Ambidexterity Levels"

_sustainability, doi:10.3390/su11185004_

Round 1

Reviewer 1 Report

These are my comments regarding the manuscript:

1) The tensions approach is based solely on Andriopoulos and Lewis (2009) Exploitation-exploration tensions and organizational ambidexterity: Managing paradoxes of innovation. Organization Science, 20(4), 696-717. However, Andriopoulos and Lewis identified the 3 dimension of tension without measuring the intensity. How you measured the intensity of each of these dimensions in your paper? The author(s) mention that “This study used a binominal scale to understand whether the tensions within organizations were balanced or unbalanced. Taking items in customer orientation as examples, Relationship orientation vs. Transaction orientation indicates two ends of the item, so the excessively concentrated relationship orientation is 1, and, on the contrary, the transaction orientation is 7”. In my opinion, this is not correct.

2) Almost all the time, when someone use cluster analysis, some of the cases does not fit in any of the clusters. However, in this case is a perfect match, all 234 responses fit in one of the four clusters.

3) Explain Growth/share factor of performance. Growth is not the same with share (I assume market share), so be more precise in explaining the factors in the model.

4) The authors rely for much of their tensions part on Andriopoulos and Lewis (2009). However, exactly their paper is not cited in the references list.

5) Table 3 title is incomplete.

6) References in the manuscript are not complying with the journal requirements.

7) Reference list does not comply with the journal requirements.

Author Response

Reviewer’s Comment 1:

The tensions approach is based solely on Andriopoulos and Lewis (2009) Exploitation-exploration tensions and organizational ambidexterity: Managing paradoxes of innovation. Organization Science, 20(4), 696-717. However, Andriopoulos and Lewis identified the 3 dimension of tension without measuring the intensity. How you measured the intensity of each of these dimensions in your paper? The author(s) mention that “This study used a binominal scale to understand whether the tensions within organizations were balanced or unbalanced. Taking items in customer orientation as examples, Relationship orientation vs. Transaction orientation indicates two ends of the item, so the excessively concentrated relationship orientation is 1, and, on the contrary, the transaction orientation is 7”. In my opinion, this is not correct. 

Author’s response:

Many thanks for the reviewer’s comment. For the measurement of organizational tension, although we adopt the definition of Andriopoulos and Lewis (2009), we can measure effectively. For example, Peng and Lin (2019) discussed the possibility of organizational tensions in Sustainability. However, in order to enhance the feasibility of this research method, relevant articles will be cited to make this measurement more reliable.

[Peng, M.Y.P.; Lin, K.H. Impact of Ambidexterity and Environmental Dynamism on Dynamic Capability Development Trade-Offs. Sustainability 2019, 11(8), 2334.] (Line 565-566)

[The nearer the score is to four, the smaller the organizational tensions will be; the nearer the score is to one or seven, the bigger the organizational tensions will be.] (Line 343-344)

Reviewer’s Comment 2:

Almost all the time, when someone use cluster analysis, some of the cases does not fit in any of the clusters. However, in this case is a perfect match, all 234 responses fit in one of the four clusters.

Author’s response:

Many thanks for the reviewer’s comment. According to the way that scholars used, we made cluster analysis step by step to divide BD and CD into four groups which cover the responses of all 234 respondents.

Reviewer’s Comment 3:

Explain Growth/share factor of performance. Growth is not the same with share (I assume market share), so be more precise in explaining the factors in the model.

Author’s response:

Many thanks for the reviewer’s comment. In this study, share means market share. In previous studies, growth and share are combined. Therefore, this may cause a difference in terms or scholars' research. However, we also agree with reviewers' comments, and make explanation for Growth/share factor of performance.

[However, scholars believe that there are some similarities between growth and market share [34], so this study classifies both together as one of the measurement variables to measure performance.] (Line 347-349)

Reviewer’s Comment 4:

The authors rely for much of their tensions part on Andriopoulos and Lewis (2009). However, exactly their paper is not cited in the references list. 

Author’s response:

Many thanks for the reviewer’s comment. This is authors’ mistake. We will add this article in reference list.

[Andriopoulos, C.; Lewis, M.W. Exploitation-exploration tensions and organizational ambidexterity: Managing paradoxes of innovation. Organization Science 2009, 20(4), 696-717.] (Line 619-620)

Reviewer’s Comment 5:

Table 3 title is incomplete. 

Author’s response:

Many thanks for the reviewer’s comment. We have completed the title of Table 3.

Reviewer’s Comment 6:

References in the manuscript are not complying with the journal requirements. 

Author’s response:

Many thanks for the reviewer’s comment. We have asked a native editor to improve our communication quality in this revised manuscript based on your suggestions. In addition, we also revised our sentences to be clearer and easier to understand.

Reviewer’s Comment 7:

Reference list does not comply with the journal requirements. 

Author’s response:

Many thanks for the reviewer’s comment. We have asked a native editor to improve our communication quality in this revised manuscript based on your suggestions. In addition, we also revised our sentences to be clearer and easier to understand.

Reviewer 2 Report

attached

Author Response

Reviewer’s Comment 1:

You should clarify the contributions of the paper which are not elaborated well in the current paper. You can talk about the following contributions: What insights can you provide based on your finding? Do they push forward our understanding? What should we do with your research? Do you have any suggestions to improve the current regulation or practice? Adding the above discussion and extend your literature review may help you make more contributions and position your contributions better. 

Author’s response:

Many thanks for the reviewer’s comment. We will explain more on contributions of the paper. We hope that reviewers and readers will have a clearer understanding of the context of this paper. In addition to adding the literature recommended by reviewer, some explanations are added to the literature review and discussion section.

[According to our findings and the above arguments, this study provides several contributions. First, this study showed the dark-side impact of DCs—influencing the operation and development of an organization means that even if organizational ambidexterity is promoted, the existence of tensions cannot be changed. For the study of dynamic capabilities, we must first understand the negative effects of tensions and then discuss the relationship between DCs and performance. Second, this study not only strengthens the description of DCs in previous studies but also integrates ambidexterity measurement methods to provide more in-depth insights. Third, although explorative and exploitative capabilities are not positively related to organizational tensions, there are still many possibilities for the combination of ambidexterity and organizational tensions in the field of DCs, which could provide a direction for future research. For example, future researchers could explore the formation of tension at the individual-level, based on the concept of corporate governance [27,36,37,59].] (Line 472-483)

Reviewer’s Comment 2:

The paper seems to claim causality but does not discuss the potential endogeneity issue and its remedies sufficiently. See Li 2016, Endogeneity in CEO power: A survey and experiment, Investment Analysts Journal, 45 (3): 149-162 for a summary of methods to deal with the endogeneity problem. No need to use all these methods but at least discuss them in your scenario.

Author’s response:

Many thanks for the reviewer’s comment. Since all the data of this study come from self-report scales, the respondents answered the degree of performance of exploration, exploitation, and organizational performance over the past three years. Therefore, this study requires the respondents to evaluate the degree of development at a specific time point; and questions are also designed in the questionnaire in order to avoid the tautology. So, this study did not conduct endogeneity tests for the measurement variables. However, the authors agree with the reviewer’s doubts. In addition to examining the size of R2, a new field was added under each model to indicate the Max VIF in the model. In the model of this study, the values from 0.043 to 0.488 indicate a relatively small proportion of residual, which means the degree of influence on the research results may be very small even if there are endogeneity problems. Besides, the value of VIF is only up to 3.834, which is below 10, indicating that the regression model does not have multicollinearity. According to reviewer’s suggestion, we explained more about potential endogeneity issue in 4.2 Hypotheses testing.

[Li [57] pointed out in the study that performance in different periods may be influenced by dependent variables, which may further generate potential endogeneity issues. However, this study required the respondents to evaluate the degree of development at a specific time point, and the questions were also designed in the questionnaire to avoid a tautology. Thus, this study did not conduct endogeneity tests for the measurement variables.] (Line 378-382)

Reviewer’s Comment 3:

Related to the above point, you should study and rationalize the use of firm size measures in the literature since frim size is the key variable in this area and they affect the independent and dependent variables simultaneously. See Dang et al. 2018. Measuring Firm Size in Empirical Corporate Finance. Journal of Banking & Finance, 86:159-176. After all it is the most significant variable in most studies alike. You need to discuss and justify your firm size measure. The results may not be robust to different measures of firm size, which is very common in this area.

Author’s response:

Many thanks for the reviewer’s comment. We will add this paper in 3.2 Measuresand make a clearer explanation for measurement of firm size.

[Based on Dang and Li’s [55] arguments, firm size is a more important fundamental firm characteristic than other control variables. Therefore, it is more likely that a correlation exists between firm size to firm performance. To manage this fixed effect, this study uses the number of employees to measure firm size.] (Line 352-355)

Reviewer’s Comment 4:

My main suggestion is that you should tell a richer story and link to more literature by discussing more relevant channels. One main channel is corporate governance. You should consider, for example, market competition as a governance mechanism:                                    X Giroud & HM Mueller 2011, Corporate governance, product market competition, and equity prices. Journal of Finance 66, 563- 600. The interactions between the executives, such as mutual monitoring among the executives: Li, Z.F., 2014, Mutual monitoring and corporate governance, Journal of Banking & Finance, 45, 255- 269; Li, Z.F., 2018, Mutual monitoring and agency problem. https://www.researchgate.net/publication/272305464_Mutual_Monitoring_and_Agency_Problems; and external interactions between CEOs in the industry tournament: Coles et al. 2018, Industry Tournament Incentives, Review of Financial Studies, 31(4):1418-1459; On inside debt as governance: Li, F., Lin, S., Sun, S., Tucker, A. 2018. Risk-Adjusted Inside Debt. Global Finance Journal 35: 12-42. Or compensation incentives: Core, J. and Guay W., 1999, The use of equity grants to manage optimal equity incentive levels, Journal of Accounting and Economics 28, 151-184. You need to discuss those aspects of possible channels to give readers a more comprehensive view and a richer story and/or point out future research direction from these perspectives.

Author’s response:

Many thanks for the reviewer’s comment. We added the concept and description of corporate governance to the hypotheses and conclusion according to the recommendations of the review committee to make the hypothesis and the theoretical framework clearer and richer.

[On the other hand, promoting performance growth by enhancing one’s dynamic capabilities is a good practice in market competition. Giroud and Mueller [27] studied this issue from the perspective of corporate governance, pointing out that the fact that firms with good corporate governance have better performance on average is well established.] (Line 57-61)

[In addition, Li [36,37] discussed the role of mutual monitoring in corporate governance. Li pointed out that information asymmetry exists in the organization among senior managers, who may have different understandings of resource input of competence and generate different tensions derived from the authority, credibility, and influence of team members.] (Line 139-143)

[Ambidexterity can also be considered good corporate governance. Li [36,37] indicated that mutual monitoring may arise in the normal methods associated with effective management.] (Line 216-218)

Reviewer’s Comment 5:

Try to use present tense throughout the paper. For example, the results indicate that…

The title reads awkward. Suggest “Dynamic capability and firm performance: ………” There are many typos and grammatical mistakes throughout the paper, making it hard to read and understand. Try to avoid long sentences and vague words. Use short, precise, and concise sentences and be more straightforward. The last section should be called conclusion (different from discussion of the results) where you should summarize all your findings, their implications to researchers and practitioners, future direction for research, limitation of the current study, etc. You need to seriously proofread the paper and extend and update your references.

In conclusion, I would like to thank the authors for a very interesting, unique and potentially important paper. Hope these comments and suggestions can help further their study.

Author’s response:

Many thanks for the reviewer’s comment. We have asked a native editor to improve our communication quality in this revised manuscript based on your suggestions. In addition, we also revised our sentences to be clearer and easier to understand.

Round 2

Reviewer 1 Report

The authors followed my recommendations.

Reviewer 2 Report

The paper is significantly improved. I suggest the authors to proofread the paper and make sure the writing meet the high standard of the journal and the logic flows smoothly.

This manuscript is a resubmission of an earlier submission. The following is a list of the peer review reports and author responses from that submission.

Round 1

Reviewer 1 Report

Dear Author,

After reviewing the paper entitled Does Dynamic Capability Matter Higher Performance in High-Tech Industry? The Effects of Tensions and Ambidexterity, I found significant text overlap with another paper, entitled Impact of Ambidexterity and Environmental Dynamism on Dynamic Capability Development Trade-Offs.

The overlap does not only concern sections like Literature review, but, most important, covers the reported results and the conceptual model.

Under these circumstances,  my recommendation is not favorable.